# Test-Time Scaling in Clinical Decision Making

**Ji Young Byun**[1,2]                                                     JBYUN13@JHU.EDU
**Young-Jin Park**[3]                                                        YOUNGP@MIT.EDU
**Navid Azizan**[3]                                                           AZIZAN@MIT.EDU
**Rama Chellappa**[1,2]                                                    RCHELLA4@JHU.EDU

[1] *Johns Hopkins University, Baltimore, MD 21218*

[2] *Johns Hopkins University, School of Medicine, Baltimore, MD 21218*

[3] *Massachusetts Institute of Technology, Cambridge, MA 02139*

**Editors:** Accepted for publication at MIDL 2026

## Abstract

Large language models (LLMs) have demonstrated remarkable capabilities in complex reasoning and knowledge-intensive tasks, yet their potential for clinical decision making through test-time scaling (TTS) remains largely unexplored. While TTS has shown promise in improving reasoning performance by leveraging additional inference-time computation, its effectiveness in the medical domain has not been systematically investigated. This gap is further exacerbated by the impracticality of supervised fine-tuning for clinical reasoning tasks, owing to limited data availability and high annotation costs. In this work, we present a comprehensive study of TTS for clinical decision making. We systematically investigate the interaction between TTS and inference strategies, including direct answering, chain-of-thought prompting, and two-stage reasoning. We generate multiple candidate outputs in parallel using large reasoning models and aggregate them via self-consistency decoding. This approach does not need any supervision while it leverages additional inference-time computation to improve the performance. We provide a comprehensive empirical evaluation across both text-based medical question answering benchmarks and medical imaging modalities, demonstrating consistent improvements over single-pass inference baselines with performance gains of up to 30 percentage points. Finally, we provide an analytical characterization of TTS, deriving scaling laws that describe how performance improves with the number of samples and identifying conditions under which TTS yields reliable gains, along with empirical validation on diverse medical decision-making tasks.

**Keywords:** Medical Image Diagnosis, Large Reasoning Model, Vision Language Model, Test-time Scaling

## 1. Introduction

Large language models (LLMs) and vision-language models (VLMs) show strong performance across diverse domains, including mathematics (Snell et al., 2024), robotics (Li et al., 2023a), autonomous driving (Qian et al., 2024), and scientific research (Xu et al., 2023; Roberts et al., 2024). A central factor in these advances is their ability to perform *reasoning*. While conventional deep learning models are often treated as black-box predictors, outputting a final result without rationale, recent large reasoning models can articulate intermediate steps explaining how answers are derived.

Explicit multi-step reasoning methods, such as chain-of-thought (CoT) prompting (Wei et al., 2022; Temsah et al., 2024), produce step-by-step explanations that enhance problem-solving in domains like arithmetic and symbolic reasoning. Early studies have applied

CoT-style prompting to medical tasks (Liu et al., 2024; Tu et al., 2025), aligning with clinical practice where physicians sequentially observe, interpret, and diagnose. However, the effectiveness of multi-stage reasoning typically depends on fine-tuning with large collections of annotated reasoning processes. In medicine, such annotations require domain experts and are costly to obtain, making supervised approaches impractical. This scarcity motivates approaches that *do not rely on fine-tuning*, which are promising for extending large reasoning models (LRMs) to medical domains without extensive supervision.

Zero-shot prediction with LRMs, however, often yields suboptimal performance. To address this, *test-time scaling* (TTS) has recently emerged as a promising inference paradigm. The key idea is to allocate additional computation during inference to improve a model's reasoning ability. A common strategy is "parallel thinking", where multiple candidate outputs are sampled and aggregated, rather than relying on a single generated output (i.e., single-pass decoding) (Yao et al., 2023; Snell et al., 2024). These approaches, ranging from majority voting (Wang et al., 2022) to verifier-based selection (Cobbe et al., 2021; Uesato et al., 2022; Wang et al., 2024b; Lightman et al., 2023), have demonstrated strong performance in domains requiring complex reasoning, such as mathematics.

Directly transferring TTS methods, particularly those leveraging verifiers, to medical applications presents significant challenges. Verifiers, known as reward models, are often unavailable in the medical domain because their training requires vast amounts of labeled reward data. For example, Qwen-PRM (Zhang et al., 2025), a reward model used for mathematical reasoning, required 4.5 million labels for its training. Therefore, TTS in medicine has focused on reward-free inference schemes, such as self-consistency decoding (Singhal et al., 2025), mostly on textual benchmarks.

Despite these efforts, our understanding of TTS in medicine remains limited. Key open questions include: **can these methods extend beyond text to multimodal medical decision making tasks?** and **under what conditions does TTS improve performance?** Motivated by these limitations, this work presents a comprehensive investigation of TTS for medical decision-making tasks, introducing a framework that enhances performance without requiring additional supervision or specialized reward models.

Our key contributions are summarized as follows:

1. We investigate inference strategies (direct answering and CoT) and introduce a two-stage reasoning framework for medical decision making, where a VLM produces textual descriptions that are aggregated by an LLM for diagnosis.

2. We evaluate TTS on test-time inference strategies and show consistent improvements, with gains of up to 30.4 percentage points over single-pass baselines.

3. We present a characterization of TTS, deriving scaling laws that describe how performance improves with the number of samples and identifying conditions under which TTS yields reliable gains.

## 2. Related Work

### 2.1. Vision-Language Models in Medical Imaging

Conventional data-driven deep learning approaches parameterize a model with learnable parameters and train it on a dataset of image–label pairs. Such models are often treated

as *black-box* predictors: they provide a final classification result but lack transparency, and the reasoning underlying the diagnostic process is non-interpretable. Given the safety-critical nature of medicine and the risk that generated content may deviate from clinical standards, rigorous evaluation is mandated to assess progress and mitigate harms (Johnson et al., 2023). Addressing these interpretability and reliability concerns is paramount for the responsible deployment of AI in clinical settings.

On the other hand, the development of VLMs has rapidly progressed, enabling models to process both images and text for diverse applications, including robotics (Li et al., 2023a), autonomous driving (Qian et al., 2024), and scientific research (Xu et al., 2023; Roberts et al., 2024). In the medical domain, early efforts focused on foundational tasks such as medical image captioning and VQA on datasets like VQA-RAD (Lau et al., 2018). More recently, models like Med-PaLM (Tu et al., 2024), Med-Flamingo (Moor et al., 2023), and LLaVA-Med (Li et al., 2023b) have demonstrated strong performance in generating clinically-relevant text. These efforts are now being pushed further by more recent works such as VILA-M3 (Nath et al., 2025) and MedXpertQA (Zuo et al., 2025), which focus on more complex reasoning and comprehensive evaluations.

The use of multi-stage reasoning, a paradigm that breaks down a complex task into a series of intermediate steps, has gained significant traction as an alternative to end-to-end approaches. This aligns with the clinical workflow, where a clinician first observes an image and other patient information, analyzes the symptoms, and then formulates a diagnosis based on their observations and knowledge. For instance, recent works have explicitly incorporated multi-stage reasoning, such as CoT (Liu et al., 2024; Tu et al., 2025), to generate detailed diagnostic rationales and explain their decision-making process. More advanced methods have also emerged, including Tree-of-Thought (ToT) (Yao et al., 2023), which creates a tree-like structure of potential diagnostic paths and evidence. This allows the model to explore and evaluate multiple hypotheses simultaneously before reaching a final conclusion.

### 2.2. Test-Time Compute Scaling

TTS has become a prominent research area, offering a computationally efficient alternative to traditional retraining for enhancing model performance. By leveraging an increased computational budget at inference time, these strategies improve a model's accuracy and robustness without requiring any changes to its parameters or architecture. CoT (Wei et al., 2022; Temsah et al., 2024) is a notable example, where a model is prompted to generate a series of intermediate reasoning steps before arriving at the final answer. While effective, CoT can be sensitive to prompting and may not always yield consistent results. TTS, a related but distinct paradigm, further improves performance by moving beyond a single, deterministic output. Instead, TTS methods sample multiple candidate outputs and aggregate them to form a more robust and reliable final prediction.

A variety of TTS strategies have been explored, ranging from simple aggregation to more complex reasoning-based methods. Simple approaches like self-consistency (Wang et al., 2022) and majority voting rely on aggregating multiple generated outputs to improve reliability. More advanced techniques have significantly pushed performance boundaries on complex benchmarks. For instance, self-refinement (Qu et al., 2024; Madaan et al., 2023) is

an iterative approach where a model critiques its own output and then revises it in a feedback loop. Similarly, verifier-based methods (Cobbe et al., 2021; Uesato et al., 2022) and process reward models (Wang et al., 2024b) have achieved state-of-the-art results by training a separate model to select the best output. Recent works have validated these approaches on increasingly challenging benchmarks, such as MATH (Hendrycks et al., 2021), GSM8K (Cobbe et al., 2021), and the BIG-Bench Hard suite (Srivastava et al., 2023), demonstrating their strong performance in mathematical and symbolic reasoning tasks.

Although powerful AI techniques are promising, their application in medicine is still emerging. One direction for improving model performance is scaling "*deep thinking*" by increasing a model's computational budget for a single reasoning path, such as by expanding its token limit (Huang et al., 2025). However, this approach faces significant challenges: it can lead to overthinking (Yang et al., 2025). Consequently, "*parallel thinking*" strategies represent another important yet unexplored avenue in the medical domain. A key barrier to those advanced methods (e.g., Best-of-$N$, and beam search (Snell et al., 2024)) is their reliance on reward models, which are often unavailable in medicine as they demand vast amounts of labeled data for training [1]. To this end, this paper explores the application of a reward-free TTS to medical image diagnosis by extending a majority voting strategy into a probabilistic framework that improves reliability.

## 3. Methods

This paper presents an approach to modality-agnostic medical decision making with the goal of generating accurate answers to clinically relevant questions based on input data, without requiring task-specific fine-tuning. Formally, a medical decision making problem (including textual question answering (QA) and medical image diagnosis) can be represented as a triplet $(\boldsymbol{x}, q, y)$, where:

- $\boldsymbol{x}$ denotes the *context*, which may take different modalities.

- $q$ represents the *problem description*, expressed as a natural language query about the context.

- $y$ is the *ground truth answer*, serving as the target output for the model.

This formalization provides a unified framework that accommodates a broad spectrum of medical QA tasks, ranging from text-based multiple choice to image-based VQA tasks. For example, in a medical image diagnosis setting, the context $\boldsymbol{x}$ may correspond to a chest X-ray, the query $q$ would be a natural language prompt such as "Does this patient have pneumonia?", and the ground truth $y$ is a categorical label (e.g., 0: normal, 1: pneumonia).

### 3.1. Test-Time Inference Strategies

This section describes two widely adopted, single-step inference strategies—zero-shot direct answering and CoT prompting—and introduces a two-stage framework specifically designed

---

1. This data is not merely correct answers but expert-annotated process supervision, where a model's step-by-step reasoning is evaluated. The high cost of clinical experts' time and the inherent complexity of medical judgment make acquiring this type of data prohibitively expensive and scarce.

for medical image diagnosis that explicitly decomposes the reasoning process into two distinct phases. See Appendix B for full prompt templates. A graphical illustration of each prompting strategy is provided in Figure 1.

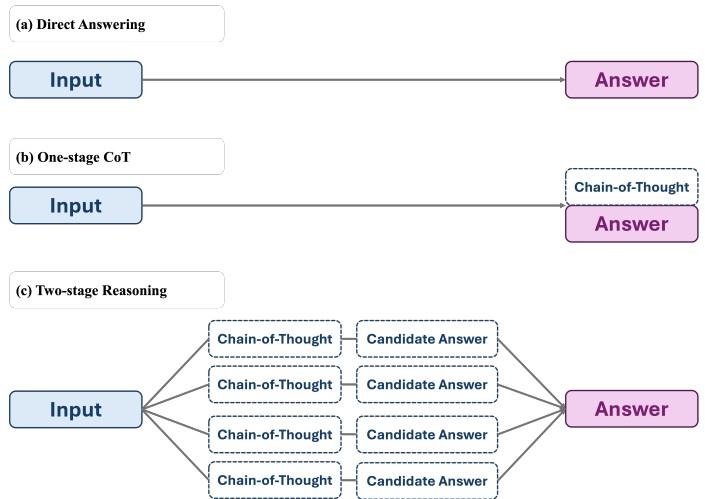

Figure 1: Comparison of three inference strategies for medical decision making. (a) *Direct Answering*: The model directly produces an answer from the input without intermediate reasoning. (b) *One-Stage CoT*: The model generates step-by-step reasoning before providing the final answer, prompted with phrases like "Let's think step by step." (c) *Two-Stage Reasoning with Self-Consistency Decoding*: For medical image diagnosis, the vision-language model first generates multiple visual descriptions of the input image, then produces candidate answers for each description. These candidate answers are aggregated via self-consistency decoding to produce the final diagnosis.

### 3.1.1. DIRECT ANSWERING

Consider a language model (LM), such as Llama 3.2-Vision instruction-tuned model, that takes a textual prompt $q$ and a context $\boldsymbol{x}$ and produces an answer $a$. A straightforward way to use such a model for diagnosis is to directly request a prediction of the target categorical label. For instance, we can set $q \leftarrow$ "Given a pediatric chest X-ray image, classify it as 0 (normal) or 1 (pneumonia)." The model directly provides a final answer without requiring any reasoning. We refer to this as the *Direct Answering* method.

### 3.1.2. CHAIN-OF-THOUGHT (CoT) PROMPTING

Alternatively, to elicit *explicit* reasoning, we employ Chain-of-Thought (CoT) prompting. While we retain the same input prompt $q$ used in the Direct Answering method, we enforce a reasoning process by introducing the trigger phrase "Let's think step by step." as a prefix to the generation (Kojima et al., 2022).

By leveraging the instruction-tuned nature of these reasoning models, this prefix acts as a directive that strictly orders the output: the model first generates intermediate reasoning steps (e.g., detailed clinical observations) and subsequently produces the final answer $a$. This ensures that the diagnosis is grounded in the generated rationale rather than being a direct, unexplained prediction. We denote this approach as the *one-stage CoT* method.

### 3.1.3. Two-Stage Reasoning for Medical Image Diagnosis

According to recent theoretical and experimental evidence (Abbe et al., 2025), the Transformer architecture often benefits when a complex task is decomposed into sub-tasks. Motivated by this observation, we propose and investigate a two-stage approach to help the Transformer arrive at a more accurate diagnosis in a medical image diagnosis setting.

**Visual Description Generation.** We first instruct the VLM to generate descriptions on visual features of the input image without directly querying for a diagnosis. Concretely, we prompt the VLM as follows: $v = \mathsf{VLM}(\boldsymbol{x}, q_1)$ where $q_1$ can be `"Describe visual features detected in the image"`.

**Diagnosis from Descriptions.** The generated visual descriptions $v$ are then provided as input to a (potentially different) LLM that produces a final diagnosis. For example, we can construct the query: $q_2(v) :=$ "`Decide which class best matches the visual features described: 0 (normal) or 1 (pneumonia). **Features:** {features}`", where we substitute `{features}` with the previously generated $v$. The corresponding diagnosis is characterized as:

$$a = \mathsf{LLM}\big(q_2(v)\big) = \mathsf{LLM}\bigg( \underbrace{\underbrace{\mathsf{VLM}(\boldsymbol{x}, q_1)}_{\text{Describe}}, q_2}_{\text{Diagnose}} \bigg) \tag{1}$$

### 3.2. Scaling Test-Time Compute

General-purpose language models such as Llama or DeepSeek often struggle to provide accurate answers in complex medical tasks, and fine-tuning is prohibitively expensive due to the scarcity of expert-annotated reasoning data (Berger et al., 2025; Naliyatthaliyazchayil et al., 2025). To address the issue, we investigate the applicability of the TTS technique—often applied in mathematical reasoning tasks (Yao et al., 2023; Snell et al., 2024). In particular, we adopt self-consistency decoding (Wang et al., 2022) given the absence of reliable reward models in medicine.

**One-Stage TTS.** We estimate class probabilities by sampling $N$ independent outputs from a large reasoning model under randomized decoding. Specifically, instead of greedy decoding which deterministically selects the next token with the highest probability, we use temperature sampling with $T = 0.7$ (Wang et al., 2022). At each token position, the model's output logits $\mathbf{z} = (z_1, \ldots, z_V)$ are converted to a probability distribution via the softmax: $p_i = \frac{\exp(z_i/T)}{\sum_{j=1}^{V} \exp(z_j/T)}$ where $V$ is the vocabulary size. The next token is then sampled from this distribution, allowing us to generate $N$ diverse reasoning paths.

Let the label space be $\mathcal{Y} = \{1, \ldots, C\}$. For each draw $i \in \{1, \ldots, N\}$, the model produces an answer string $a^{(i)}$, which we map to a class via a parsing function $\phi : \text{text} \to \mathcal{Y}$

(e.g., extracting "A/B/C/D" or $\{1, \ldots, C\}$). Denote the parsed class by $\hat{a}^{(i)} = \phi(a^{(i)}) \in \mathcal{Y}$. Formally,

$$\{a^{(i)}\}_{i=1}^{N} \overset{\text{i.i.d.}}{\sim} \mathsf{LRM}(\boldsymbol{x}, q), \qquad \hat{y}^{(i)} = \phi(a^{(i)}).$$

Each $\hat{y}^{(i)}$ can be viewed as a draw from the LM-induced predictive distribution over classes, $p(y \mid \boldsymbol{x}, q)$. We estimate these class probabilities by Monte Carlo:

$$\widehat{p}(y = c \mid \boldsymbol{x}, q) \;=\; \frac{1}{N} \sum_{i=1}^{N} \mathbb{I}(\hat{y}^{(i)} = c). \tag{2}$$

The final prediction is the maximum-probability class under this estimate:

$$\hat{y} \;=\; \arg\max_{c \in \{1, \ldots, C\}} \widehat{p}(y = c \mid \boldsymbol{x}, q). \tag{3}$$

**Two-Stage TTS.** For two-stage inference in medical image diagnosis, we can apply TTS both in the description stage and in the diagnosis stage. Formally,

$$\{v^{(i)}\}_{i=1}^{N} \overset{\text{i.i.d.}}{\sim} \mathsf{VLM}(\boldsymbol{x}, q_1) \tag{4}$$

$$\{a^{(i,j)}\}_{j=1}^{M} \overset{\text{i.i.d.}}{\sim} \mathsf{LRM}(v^{(i)}, q_2). \tag{5}$$

where $v^{(i)}$ denotes the $i$-th description sampled from the VLM in the first stage, and $a^{(i,j)}$ is the $j$-th diagnosis from the language model given that description in the second stage.

Empirically, we observe that even under randomized decoding, the diagnosis $a^{(i,j)}$ remains unchanged for a fixed description $v^{(i)}$. This indicates that the predictive uncertainty originates from the reasoning process (description stage) rather than from the decision-making process (diagnosis stage). Consequently, there is no measurable gain from scaling test-time compute in the second stage, and we therefore set $M = 1$. The final class probabilities and prediction are then estimated in the same way as in the single-stage case.

## 4. Results and Discussion

### 4.1. Datasets and Models

As an initial proof of concept, we first evaluate TTS on text-based medical QA tasks using the Massive Multitask Language Understanding (MMLU) benchmark (Hendrycks et al., 2020). We focus on six medically relevant domains: clinical knowledge, medical genetics, anatomy, professional medicine, college biology, and college medicine. Since these are multiple-choice questions, all answer options are included in the prompt along with the question. The detailed prompts and inference setting are provided in the Appendix B.

To further assess generalizability across modalities and disease types in medical image diagnosis, we use PneumoniaMNIST, PathMNIST, and RetinaMNIST from MedMNIST v2 (Yang et al., 2023). Specifically, pneumonia detection is performed using PneumoniaMNIST, which consists of 390 pneumonia cases and 234 normal cases from frontal X-ray images. PathMNIST is utilized for colorectal cancer classification, containing 1,233 cases of colorectal adenocarcinoma epithelium and 741 cases of normal colon mucosa. Diabetic

| Method | Clinical Knowledge | Medical Genetics | Anatomy | Professional Medicine | College Biology | College Medicine |
|---|---|---|---|---|---|---|
| *Llama-3.1-8B-Instruct* | | | | | | |
| Direct Answering | 0.71 | 0.75 | 0.62 | 0.73 | 0.75 | 0.64 |
| Direct Answering (+**TTS**) | 0.72 (↑1.3pp) | 0.78 (↑3.5pp) | 0.65 (↑2.8pp) | 0.77 (↑3.6pp) | 0.77 (↑2.3pp) | 0.67 (↑3.1pp) |
| One-stage CoT | 0.71 | 0.77 | 0.66 | 0.71 | 0.73 | 0.65 |
| One-stage CoT (+**TTS**) | **0.80** (↑9.2pp) | **0.84** (↑7.6pp) | **0.72** (↑5.7pp) | **0.85** (↑14.3pp) | **0.84** (↑11.2pp) | **0.77** (↑11.8pp) |
| *DeepSeek-R1-Distill-Llama-8B* | | | | | | |
| Direct Answering | 0.52 | 0.55 | 0.48 | 0.46 | 0.54 | 0.47 |
| Direct Answering (+**TTS**) | 0.56 (↑3.8pp) | 0.62 (↑6.8pp) | 0.51 (↑3.4pp) | 0.57 (↑10.6pp) | 0.62 (↑8.1pp) | 0.52 (↑5.0pp) |
| One-stage CoT | 0.61 | 0.63 | 0.53 | 0.58 | 0.65 | 0.58 |
| One-stage CoT (+**TTS**) | **0.73** (↑12.1pp) | **0.80** (↑17.5pp) | **0.64** (↑11.3pp) | **0.72** (↑14.4pp) | **0.80** (↑15.2pp) | **0.74** (↑15.5pp) |

Table 1: Accuracy on six medical domains of MMLU using different prompting strategies. We compare **baselines** (direct answering and one-stage chain-of-thought (CoT)) with our **test-time scaling (TTS)** variants for $N = 64$. Across both Llama-3.1-8B-Instruct and DeepSeek-R1-Distill-Llama-8B, applying TTS consistently improves performance over their respective baselines, with the largest gains observed for one-stage CoT, up to 17.5 percentage points (pp).

retinopathy (DR) classification is explored with RetinaMNIST, which includes 226 cases of referrable (i.e., non-proliferative or proliferative DR) and 174 normal cases from fundus images. All images are standardized to a resolution of $224 \times 224$.

For the MMLU benchmark, we primarily evaluate Llama-3.1-8B-Instruct (Touvron et al., 2023) and DeepSeek-R1-Distill-Llama-8B (Guo et al., 2025), as well as additional results with Llama-3.2-1B-Instruct and Llama-3.2-3B-Instruct. For the medical image diagnosis, we employ Llama-3.2-11B-Vision-Instruct (Touvron et al., 2023). Since the second stage of our two-stage inference framework admits flexible model selection, we further experiment with smaller text-only Llama models (1B, 3B, and 8B) as well as the medical-domain–specific Med42-v2-8B model (Christophe et al., 2024).

### 4.2. Comparison with Baselines

We compare TTS effectiveness against conventional baselines across three test-time inference settings: (1) direct answering, (2) one-stage CoT, and (3) two-stage reasoning framework for medical image diagnosis. Each setting is assessed with and without the proposed TTS strategy.

Table 1 and Table 2 show that the TTS strategy consistently delivers substantial performance gains across diverse tasks, models, and test-time inference strategies. While one-stage CoT without TTS often yields marginal or negative gains, TTS shows strong effects on vision-centric tasks through multi-sample aggregation. Our analysis further reveals several key observations:

- **Consistent gains across tasks and models.** On the MMLU dataset (Table 1), TTS improves performance across six medical knowledge areas, up to 17.5 percentage points

| Method | Pneumonia | | Colorectal Cancer | | Diabetic Retinopathy | |
|---|---|---|---|---|---|---|
| | AUC | AP | AUC | AP | AUC | AP |
| *Llama-3.2-11B-Vision-Instruct* | | | | | | |
| Direct Answering | 0.50 | 0.62 | 0.56 | 0.65 | 0.61 | 0.63 |
| Direct Answering (+**TTS**) | 0.74 (↑+24.2pp) | 0.79 (↑+16.9pp) | 0.56 (↑+0.5pp) | 0.66 (↑+0.3pp) | **0.71** (↑+10.0pp) | 0.74 (↑+11.1pp) |
| One-stage CoT | 0.53 | 0.64 | 0.48 | 0.62 | 0.58 | 0.61 |
| One-stage CoT (+**TTS**) | 0.78 (↑+24.9pp) | 0.83 (↑+18.8pp) | 0.53 (↑+5.4pp) | 0.64 (↑+2.4pp) | 0.67 (↑+9.2pp) | **0.74** (↑+13.4pp) |
| Two-stage Reasoning | 0.52 | 0.63 | 0.54 | 0.65 | 0.57 | 0.61 |
| Two-stage Reasoning (+**TTS**) | **0.82** (↑+30.4pp) | **0.86** (↑+22.8pp) | **0.65** (↑+10.9pp) | **0.75** (↑+10.6pp) | **0.71** (↑+13.5pp) | **0.74** (↑+13.0pp) |

Table 2: Results on MedMNIST datasets: PneumoniaMNIST (pneumonia detection), PathMNIST (colorectal cancer classification), and RetinaMNIST (diabetic retinopathy detection). We compare **baselines** (direct answering, one-stage CoT, and two-stage reasoning) with their **TTS** variants for $N = 16$. Across all datasets, applying TTS yields substantial improvements, with two-stage reasoning + TTS achieving the best overall performance. Largest gains observed for two-stage reasoning, up to 30.4 percentage points (pp). Metrics reported are AUC (area under the ROC curve) and AP (area under the precision–recall curve).

(pp). These trends also hold on the more challenging MedQA benchmark (Table C.1). For medical image diagnosis (Table 2), TTS consistently boosts AUC and AP scores across modalities, with gains up to 30.4 pp. This indicates the advantage of TTS is not task- or model-specific, but generalizes across text- and vision-based medical tasks.

- **TTS outperforms CoT.** While prompt engineering alone yields marginal or unstable effects, as seen in the gap between direct answering and one-stage CoT, TTS consistently produces gains regardless of the prompting strategy. This highlights TTS as a more reliable mechanism for enhancing model performance than reformatting instructions.

- **Strong effects on vision tasks.** Our TTS strategy achieves its most pronounced improvements on vision-centric tasks (Table 2). Single-pass VLM often overlooks subtle visual cues or produces ambiguous descriptions, whereas the multi-sample nature of TTS allows diverse perspectives to be aggregated into a more reliable representation. These improvements arise from the synergy between structured reasoning and test-time scaling, with scaling serving as the key driver of robustness.

### 4.3. Scaling Laws for Test-Time Compute

We analyze TTS effectiveness by systematically varying the number of samples from $N = 1$ (i.e., single-pass inference) up to $N = 64$ for text-based MMLU benchmarks and up to $N = 16$ for vision-based diagnosis tasks. As shown in Figure 2, **performance scales monotonically with compute scale**, particularly up to medium sample sizes (e.g., $N \leq 16$): accuracy improves substantially as the model aggregates multiple complementary reasoning processes, reducing reliance on potentially flawed single explanations. More comprehensive results are reported in Figure C.1 in the Appendix.

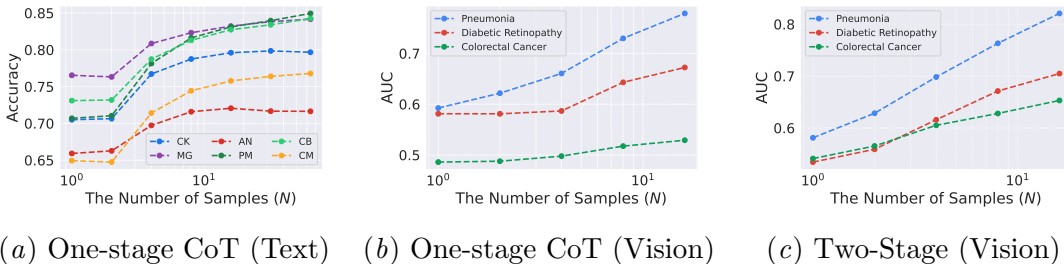

$(a)$ One-stage CoT (Text)     $(b)$ One-stage CoT (Vision)     $(c)$ Two-Stage (Vision)

Figure 2: The effect of sample size (N) in TTS setting. Increasing the sample size boosts performance across different datasets and inference methods. Llama-3.1-8B-Instruct and Llama-3.2-11B-Vision-Instruct are used for text and vision tasks, respectively.

This scaling behavior parallels our theoretical justifications in Corollary 1 in Section 5 and prior observations in Beeching et al. (2024) in language model reasoning tasks. Importantly, the observed gains in medical image diagnosis suggest such scaling laws extend beyond mathematical reasoning to multimodal medical applications.

From a practical standpoint, this result underscores a critical lesson: **relying on a single model output is unreliable in medical tasks**, as LLMs can generate plausible yet misleading information in specialized contexts[2]. In contrast, test-time compute elevates diagnostic performance up to approximately 80% without additional fine-tuning or retraining. This improvement is also practically viable: inference cost scales linearly with $N$, as prompt processing is $O(1)$ via KV caching (Pope et al., 2023) and self-consistency aggregation is negligible on CPU, meaning Figure 2 directly reflects the cost–performance trade-off. Furthermore, parallelizing generation across multiple GPUs can substantially reduce latency. This highlights TTS as a promising avenue for improving both reliability and safety in real-world medical AI deployment.

### 4.4. Model Capacity Matters for TTS

So far, we have observed that TTS consistently improves zero-shot performance and exhibits stronger synergy with models possessing sufficient reasoning ability (e.g., 8B and above). To investigate this trend, we conduct an ablation study on MMLU using smaller models (Table 3). While TTS provides modest improvements for direct answering, applying one-stage CoT substantially degrades performance in smaller models. This degradation is further amplified by TTS. For instance, with the 1B model, one-stage CoT cuts accuracy by more than half, and combining it with TTS drives accuracy to near-zero. These results highlight that TTS effectiveness depends critically on baseline model competence. When models exhibit non-trivial accuracy, TTS enhances reasoning; conversely, when models struggle to reason, scaling reinforces biased or uninformative outputs, as we formally show in Propo-

---

2. For instance, across all test-time inference methods, single-sample inference occasionally yields AUCs of only 50–60% on disease classification tasks.

| Method | Clinical Knowledge | Medical Genetics | Anatomy | Professional Medicine | College Biology | College Medicine |
|---|---|---|---|---|---|---|
| *Llama-3.2-1B-Instruct* | | | | | | |
| Direct Answering | 0.35 | 0.34 | 0.41 | 0.31 | 0.35 | 0.33 |
| Direct Answering (+**TTS**) | **0.41** (↑6.2pp) | **0.39** (↑4.7pp) | **0.47** (↑6.5pp) | **0.38** (↑6.1pp) | **0.40** (↑4.4pp) | **0.39** (↑5.8pp) |
| One-stage CoT | 0.16 | 0.16 | 0.21 | 0.15 | 0.18 | 0.17 |
| One-stage CoT (+**TTS**) | 0.04 (↓12.1pp) | 0.02 (↓14.1pp) | 0.08 (↓12.8pp) | 0.01 (↓13.9pp) | 0.05 (↓12.5pp) | 0.07 (↓9.6pp) |
| *Llama-3.2-3B-Instruct* | | | | | | |
| Direct Answering | 0.60 | 0.65 | 0.57 | 0.68 | 0.65 | 0.54 |
| Direct Answering (+**TTS**) | **0.64** (↑3.7pp) | **0.70** (↑4.4pp) | **0.64** (↑6.6pp) | **0.77** (↑8.7pp) | **0.70** (↑5.1pp) | **0.55** (↑0.8pp) |
| One-stage CoT | 0.45 | 0.49 | 0.48 | 0.39 | 0.38 | 0.38 |
| One-stage CoT (+**TTS**) | 0.57 (↑12.4pp) | 0.66 (↑17.1pp) | 0.62 (↑14.5pp) | 0.50 (↑11.2pp) | 0.45 (↑7.8pp) | 0.45 (↑7.2pp) |

Table 3: Accuracy on six medical domains of MMLU using different prompting strategies. We compare **baselines** (direct answering and one-stage chain-of-thought (CoT)) with their **TTS** variants using $N = 64$. For LLAMA-3.2-1B-INSTRUCT, CoT prompting substantially degrades performance, and applying TTS further amplifies this degradation. For LLAMA-3.2-3B-INSTRUCT, CoT lowers baseline accuracy, but TTS recovers performance, yielding consistent improvements. These results suggest TTS is most effective when models achieve non-trivial accuracy (above random guessing, i.e., ~25% for four-choice questions); otherwise, scaling may reinforce biased or uninformative reasoning.

sition 1 (Section 5). This underscores that naively introducing reasoning prompts can be counterproductive without sufficient underlying capability.

## 5. When Does TTS Help? An Analytical Justification

While self-consistency decoding has demonstrated strong empirical performance in medical applications (Singhal et al., 2023, 2025) and mathematical reasoning tasks (Beeching et al., 2024), it remains underexplored whether TTS can be applied across different LLMs and how its scaling behavior unfolds (e.g., whether it converges quickly or grows monotonically). To address this gap, we first present a theoretical analysis of TTS based on self-consistency decoding. Proofs are in Appendix A.

**Setup.** Consider a $C$-class classification with true class $c^\star$. A single decode (vote) from the LM yields label $y \in \{1, \ldots, C\}$ with

$$\mathbb{P}(y = c^\star) = p, \qquad \mathbb{P}(y = j) = p_j \ \ (j \neq c^\star) \tag{6}$$

where $p + \sum_{j \neq c^\star} p_j = 1$. We draw $N$ i.i.d. votes, let $X_j$ be the number of votes for class $j$, and predict by majority vote $\hat{y}_{\mathrm{MV}} = \arg\max_j X_j$ (break ties uniformly at random). Define the strongest competitor $q := \max_{j \neq c^\star} p_j$.

**Proposition 1 (Majority vote vs. strongest competitor)** *If $p > q$, then*

$$\mathbb{P}(\hat{y}_{\mathrm{MV}} \neq c^{\star}) \leq (C-1) \exp\Big(-\tfrac{N}{2}(p-q)^2\Big), \tag{7}$$

*so the error decays exponentially in $N$, and improves as the margin $p - q$ grows (i.e., LM becomes more confident).*

*Conversely, if $q > p$, then*

$$\mathbb{P}(\hat{y}_{\mathrm{MV}} = c^{\star}) \leq (C-1) \exp\Big(-\tfrac{N}{2}(q-p)^2\Big), \tag{8}$$

*so $\hat{y}_{\mathrm{MV}}$ amplifies the wrong class as $N$ grows.*

**Corollary 1 (Exponential scaling)** *If $p > q$, the error of $\hat{y}_{\mathrm{MV}}$ decays exponentially with $N$, and to achieve $\mathbb{P}(\hat{y}_{\mathrm{MV}} \neq c^{\star}) \leq \delta$ it suffices that*

$$N \geq \frac{2}{(p-q)^2} \log\Big(\frac{C-1}{\delta}\Big). \tag{9}$$

*If $q > p$, then $\mathbb{P}(\hat{y}_{\mathrm{MV}} = c^{\star})$ decays exponentially in $N$ at the same rate.*

**Summary of the theoretical findings.** Proposition 1 shows, if $p > q$, exponential decay of the error in $N$; if $q > p$, majority vote amplifies the wrong label. Hence: (1) self-consistent TTS improves with larger $N$ *in regimes where the true class has the largest single-pass probability*, and (2) it is *effective only when the LLM is sufficiently confident*, in the sense of a nontrivial margin $p > q$.

## 6. Conclusion

We present a comprehensive study of TTS for clinical decision making, encompassing textual QA benchmarks and medical image diagnosis with both empirical and analytical components. TTS consistently improves performance over single-pass baselines by up to 30.4 percentage points, and we provide scaling laws characterizing when such improvements emerge. Our analysis reveals TTS is most effective when underlying models possess non-trivial baseline competence, as scaling amplifies informative reasoning rather than biased outputs. Experiments confirm TTS generalizes beyond text, yielding strong improvements on vision-centric tasks and extending to domain-specific medical VLMs such as LLaVA-Med (Table C.2). Crucially, this parallel reasoning approach improves zero-shot performance without costly supervision, addressing the scarcity of high-quality medical annotations for training verifiers or reward models.

We note that our study focuses on reward-free TTS. Verifier-based methods such as best-of-N remain unexplored because reliable medical reward models do not yet exist as training them demands massive labeled data. Existing efforts remain either text-only (Wang et al., 2024a) or domain-specific to radiology reports (Thomas et al., 2025). Future directions include adaptive TTS strategies that dynamically allocate compute, developing domain-specific reward models for multimodal medical reasoning, and investigating interactions with domain-specialized models to assess clinical workflow integration and trustworthy decision-making.

## Acknowledgments

Ji Young Byun was supported in part by a discretionary fund at the Johns Hopkins Whiting School of Engineering. Young-Jin Park and Navid Azizan acknowledge support from the MIT-Amazon Science Hub, the MIT-IBM Watson AI Lab, Jane Street, and MathWorks.

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

## Appendix A. Omitted Proofs

**Proof of Proposition 1.** **Proof**

For the first case, we bound the probability of error by applying a union bound over all possible failure modes. An error occurs if at least one competitor class $j \neq c^\star$ receives at least as many votes as the true class $c^\star$.

$$\mathbb{P}(\hat{y}_{\mathrm{MV}} \neq c^\star) = \mathbb{P}\Big( \bigcup_{j \neq c^\star} \{X_j \geq X_{c^\star}\}\Big)$$
$$\leq \sum_{j \neq c^\star} \mathbb{P}(X_j \geq X_{c^\star}). \qquad (10)$$

For each competitor $j$, let $D_j := X_{c^\star} - X_j$. The term $\mathbb{P}(X_j \geq X_{c^\star})$ is equivalent to $\mathbb{P}(D_j \leq 0)$. The quantity $D_j$ is a sum of $N$ i.i.d. random variables $V_i = \mathbf{1}\{y_i = c^\star\} - \mathbf{1}\{y_i = j\}$, where each $V_i \in \{-1, 0, 1\}$. The expectation is $\mathbb{E}[D_j] = N(p - p_j)$. By Hoeffding's inequality (with variable range $1 - (-1) = 2$):

$$\mathbb{P}(D_j \leq 0) = \mathbb{P}\big(D_j - \mathbb{E}[D_j] \leq -N(p - p_j)\big)$$
$$\leq \exp\Big(-\frac{2(N(p - p_j))^2}{N \cdot 2^2}\Big) \qquad (11)$$
$$= \exp\Big(-\tfrac{N}{2}(p - p_j)^2\Big).$$

Since $q = \max_{k \neq c^\star} p_k$, we have $p - p_j \geq p - q$ for all $j \neq c^\star$. This implies $(p - p_j)^2 \geq (p - q)^2$. We can thus bound each term in the sum by the worst case:

$$\mathbb{P}(\hat{y}_{\mathrm{MV}} \neq c^\star) \leq \sum_{j \neq c^\star} \exp\Big(-\tfrac{N}{2}(p - p_j)^2\Big)$$
$$\leq \sum_{j \neq c^\star} \exp\Big(-\tfrac{N}{2}(p - q)^2\Big) \qquad (12)$$
$$= (C - 1) \exp\Big(-\tfrac{N}{2}(p - q)^2\Big).$$

For the second case, let $j^\dagger \in \arg\max_{j \neq c^\star} p_j$ so that $p_{j^\dagger} = q$. For the true class $c^\star$ to win, it must receive more votes than any other class, including the strongest competitor $j^\dagger$. Thus, the event $\{\hat{y}_{\mathrm{MV}} = c^\star\}$ is a subset of the event $\{X_{c^\star} > X_{j^\dagger}\}$.

$$\mathbb{P}(\hat{y}_{\mathrm{MV}} = c^\star) \leq \mathbb{P}(X_{c^\star} > X_{j^\dagger}). \qquad (13)$$

Let $D := X_{c^\star} - X_{j^\dagger}$. The expectation is $\mathbb{E}[D] = N(p - q) < 0$. We bound $\mathbb{P}(D > 0)$. By Hoeffding's inequality:

$$\mathbb{P}(D > 0) = \mathbb{P}\big(D - \mathbb{E}[D] > -N(p - q)\big)$$
$$= \mathbb{P}\big(D - \mathbb{E}[D] > N(q - p)\big) \qquad (14)$$
$$\leq \exp\Big(-\tfrac{N}{2}(q - p)^2\Big).$$

$\blacksquare$

## Appendix B. Prompts

For evaluation on the MMLU benchmark, we employed two primary prompt formats: direct answering and one-stage Chain-of-Thought (CoT). For medical image diagnosis on the MedMNIST dataset, we employed three prompt formats: direct answering, one-stage Chain-of-Thought (CoT), and our proposed two-stage reasoning.

### B.1. Prompt Structure for MMLU Evaluation

#### B.1.1. DIRECT ANSWERING PROMPT

In the direct answering prompt, the model is instructed to select the correct letter choice without providing any intermediate explanation or reasoning. This setup evaluates the model's immediate knowledge of the subject matter. An example direct answering prompt is shown in the visualization below.

> **User Input**
>
> The following are multiple-choice questions (with answers) about {subject}. Provide your answer with "The answer is (X)" where X is the correct letter choice, with no additional explanation.
>
> **Question:** {question}
>
> **Options:** A. {o1}, B. {o2}, C. {o3}, D. {o4}

#### B.1.2. CHAIN-OF-THOUGHT (COT) PROMPT

The one-stage Chain-of-Thought (CoT) prompt instructs the model to produce step-by-step reasoning before selecting a final answer. We implement this by explicitly asking the model to "think step by step," then report the final letter choice. This setting assesses the model's reasoning ability in addition to its factual knowledge.

> **User Input**
>
> The following are multiple-choice questions (with answers) about {subject}. Think step by step and then finish your answer with "The answer is (X)" where X is the correct letter choice.
>
> **Question:** {question}
>
> **Options:** A. {o1}, B. {o2}, C. {o3}, D. {o4}

> **Assistant Response (Prefix)**
>
> **Answer:** Let's think step by step.

## B.2. One-Stage Prompt Structure for MedMNIST

In medical imaging tasks with one-stage inference (i.e., direct answering and one-stage CoT), we use a direct-instruction format: the model receives a single-turn system prompt that specifies the classification task and the required output format. For one-stage CoT, we append the cue "Let's think step by step." to the prompt.

### B.2.1. PNEUMONIA DETECTION

**User Input**

Your task is binary-class classification of 'pneumonia' against 'normal'. Given a grayscale pediatric chest X-ray image, classify it as 0 (normal) or 1 (pneumonia). Make sure to put the answer (and only answer) inside \boxed{}.

### B.2.2. COLORECTAL CANCER

**User Input**

Your task is binary-class classification of 'malignant: colorectal adenocarcinoma epithelium' against 'normal'. Given a hematoxylin & eosin stained histological image, classify it as 0 (normal) or 1 (malignant). Make sure to put the answer (and only answer) inside \boxed{}.

### B.2.3. DIABETIC RETINOPATHY

**User Input**

Your task is binary-class classification of 'diabetic retinopathy (DR)' against 'normal'. Given a retina fundus image, classify it as 0 (normal) or 1 (DR). Make sure to put the answer (and only answer) inside \boxed{}.

## B.3. Two-Stage Prompt Structure for MedMNIST

In the two-stage reasoning setup for medical image diagnosis, the prompt is structured into two phases: (1) a general instruction asking the model to summarize the visual features from a given image, and (2) a task-specific prompt that provides detailed guidelines and questions, combined with the summary generated in the first stage (referred to as the note).

### B.3.1. STAGE 1 PROMPT FOR ALL TASKS

**User Input**

Summarize the list of key observable features detected in the image using bullet points.

B.3.2. Stage 2 Prompt for Pneumonia Detection

---

**User Input**

You are a healthcare professional to provide accurate pneumonia diagnosis.
Task:
- You will receive a report describing a patient's pediatric chest X-Ray image.
- Your goal is to classify:
- 0 = normal
- 1 = pneumonia

Guidelines:
1. Carefully read the note.
2. Decide which class (0 or 1) best matches the clinical features described. Assume that all of the relevant details have been explained in the text.
3. Provide your final answer enclosed in \boxed{} with no additional explanation, e.g., \boxed{1}.

IMPORTANT:
- Strictly adhere to the format by outputting only the final grade inside \boxed{} and nothing else.

**Note:**
{note}

—

**Question:**
Based on the above note, what is the correct pneumonia diagnosis? Please consider that all necessary details have been provided in the text above. Remember to provide only the class (0 or 1) inside \boxed{}.

---

### B.3.3. Stage 2 Prompt for Colorectal Cancer

---

**User Input**

You are a pathologist to provide an accurate colorectal adenocarcinoma epithelium diagnosis.
Task:
- You will receive a report describing a patient's hematoxylin & eosin stained histological image.
- Your goal is to classify the tissue type:
- 0 = normal
- 1 = malignant (colorectal adenocarcinoma epithelium)

Guidelines:
1. Carefully read the note.
2. Decide which class (0 or 1) best matches the clinical features described. Assume that all of the relevant details have been explained in the text.
3. Provide your final answer enclosed in \boxed{} with no additional explanation, e.g., \boxed{1}.

IMPORTANT:
- Strictly adhere to the format by outputting only the final grade inside \boxed{} and nothing else.
**Note:**
{note}

—

**Question:**
Based on the above note, what is the correct tissue type? Please consider that all necessary details have been provided in the text above. Remember to provide only the class (0 or 1) inside \boxed{}.

---

### B.3.4. Stage 2 Prompt for Diabetic Retinopathy

---

**User Input**

You are an ophthalmologist to provide accurate diabetic retinopathy (DR) diagnosis.
Task:
- You will receive a report describing a patient's retina fundus image.
- Your goal is to classify:
- 0 = normal
- 1 = referrable

Guidelines:
1. Carefully read the note.
2. Decide which class (0 or 1) best matches the clinical features described. Assume that all of the relevant details have been explained in the text.
3. Provide your final answer enclosed in \boxed{} with no additional explanation, e.g., \boxed{1}.

IMPORTANT:
- Strictly adhere to the format by outputting only the final grade inside \boxed{} and nothing else.

**Note:**
{note}

—

**Question:**
Based on the above note, what is the correct diabetic retinopathy (DR) diagnosis? Please consider that all necessary details have been provided in the text above. Remember to provide only the class (0 or 1) inside \boxed{}.

## Appendix C. Additional Experiments

| Method | $N = 1$ | $N = 4$ | $N = 16$ | $N = 64$ |
|---|---|---|---|---|
| *Llama-3.1-8B-Instruct* | | | | |
| Direct Answering | 58.0±4.7 | 60.3±0.8 | 61.7±0.5 | **61.9±0.3** |
| One-stage CoT | 62.7±1.5 | 69.9±0.9 | 74.6±0.6 | **75.6±0.3** |
| *Llama-3.2-3B-Instruct* | | | | |
| Direct Answering | 47.7±4.9 | 50.1±0.8 | 51.2±0.6 | **51.9±0.3** |
| One-stage CoT | 37.9±2.3 | 42.9±1.0 | 49.6±0.9 | **52.9±0.5** |
| *Llama-3.2-1B-Instruct* | | | | |
| Direct Answering | 29.7±3.4 | 31.3±1.1 | 33.4±0.8 | **34.8±0.5** |
| One-stage CoT | **16.1±1.3** | 12.6±1.0 | 6.4±0.5 | 3.1±0.3 |
| *DeepSeek-R1-Distill-Llama-8B* | | | | |
| Direct Answering | 37.3±5.0 | 40.1±0.9 | 42.3±0.6 | **42.7±0.4** |
| One-stage CoT | 50.2±1.1 | 55.1±0.9 | 57.6±0.7 | **58.2±0.4** |

Table C.1: MedQA with Self-Consistency TTS. Accuracy (%) is reported as mean ± standard deviation across multiple runs. Bold indicates the best performance for each model-prompt combination. TTS consistently improves performance with increasing $N$, except for LLAMA-3.2-1B-INSTRUCT with CoT, where scaling amplifies degraded reasoning.

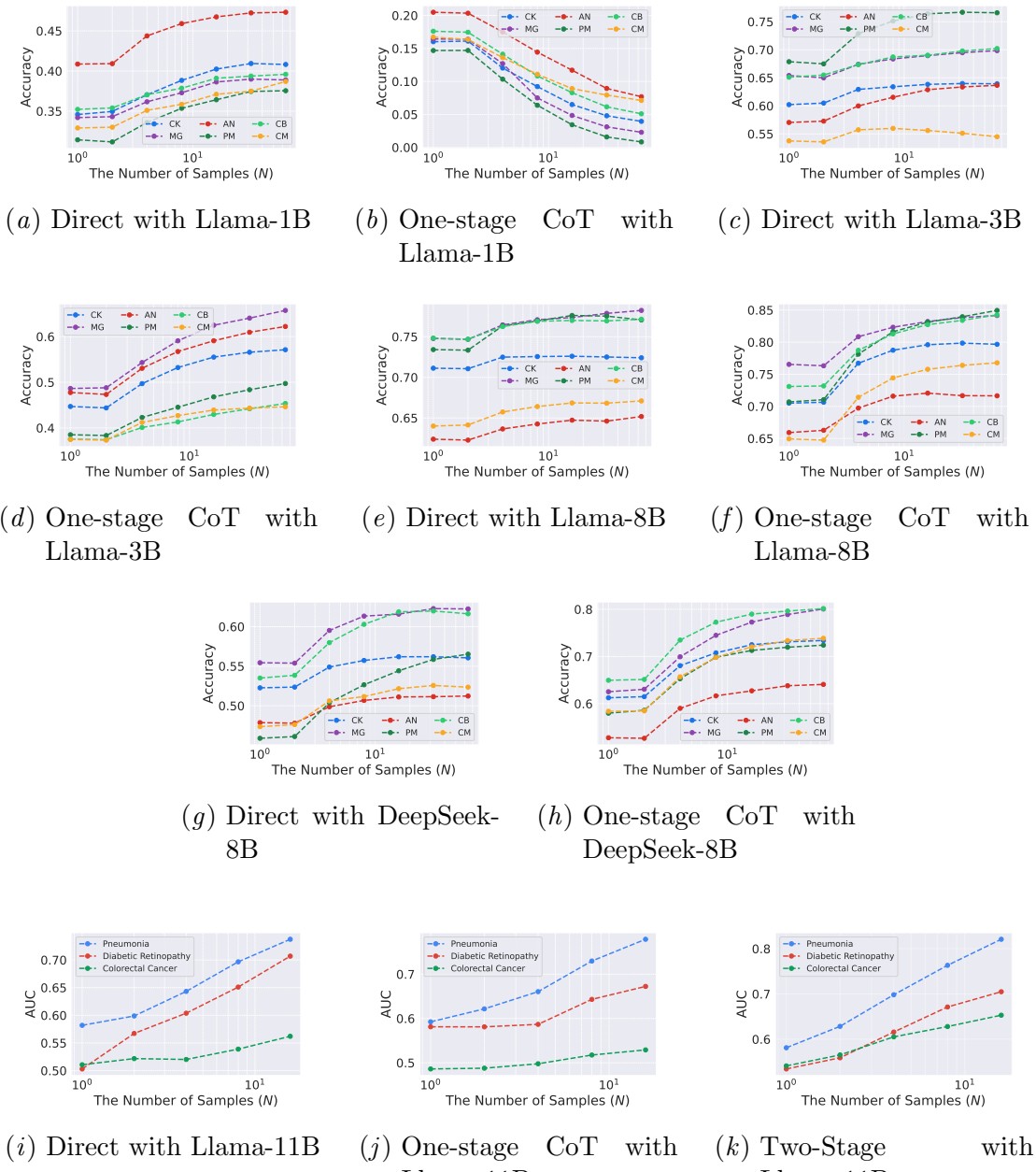

Figure C.1: A study examining the effect of sample size (N) in TTS setting. (a)-(h): text dataset. (i)-(k): vision dataset. Increasing the sample size boosts performance across different datasets and inference methods.

| Method | SLAKE | RAD-VQA |
|---|---|---|
| *LLaVA-Med-v1.5-Mistral-7B* | | |
| Direct Answering | 0.52 | 0.59 |
| Direct Answering (**+TTS**) | 0.53 (↑0.9pp) | 0.58 (↓0.8pp) |
| One-stage CoT | 0.54 | 0.54 |
| One-stage CoT (**+TTS**) | 0.55 (↑1.7pp) | **0.59** (↑5.6pp) |
| Two-stage Reasoning | 0.65 | 0.57 |
| Two-stage Reasoning (**+TTS**) | **0.66** (↑2.0pp) | **0.59** (↑2.0pp) |

Table C.2: Results on SLAKE and RAD-VQA using LLaVA-Med-v1.5-Mistral-7B with $N = 16$. Two-stage reasoning achieves the best overall performance, and TTS provides consistent improvements across inference strategies.

