# OpenReview forum: "Test-Time Scaling in Clinical Decision Making"
_MIDL.io/2026/Conference — MIDL 2026 Poster_

### Official Review · Reviewer_xniB · 2026-01-01

**Confidence:** 5
**Preliminary Rating:** 4
**Final Rating:** 4

**Summary:**

The paper studies test-time scaling (TTS) for clinical decision making by exploiting the “reasoning” capabilities of large models. Given the challenges of obtaining substantial collections of reasoning examples in medicine, the goal is to avoid supervised or reward-based approaches that guide models in the reasoning process. The authors follow the trends in TTS, propose a two-stage mechanism for reasoning and study the effects of scaling both in terms of test-time compute and model capacity.

**Strengths:**

It is a timely and interesting study to translate the developments in TTS to medical applications and seek clever ways to make the most out of large models, both in vision and language domains. It comes with a decent methodology and evaluation, and offers insights into the impact of TTS and model capacity. A particular example is the finding that TTS boosts performance when used in tandem with large models, but this is not the case with small models. This reminds me of the rule of thumb with ensemble learning: model diversity is key for obtaining high performance. Probably, small models tend to be less diverse.

Another interesting finding is the benefits of TTS, when CoT falls short of improvements, especially when a small model with limited capacity is used. It would be also easier to use TTS instead of writing out detailed reasoning prompts in medicine at scale.

**Weaknesses:**

The manuscript is in decent shape; however, it gives me the impression that it was written as a full-blown article and then compressed into a conference version with some crucial components buried into the appendix, especially the related work. While this is a common approach for venues in life sciences and medicine, which tend to sidestep technicalities and highlight findings, I find it at odds with the needs of more technical venues, such MIDL. I believe readers with technical focus would benefit from those parts and love to see them in the main text.

**Detailed Comments:**

Towards the end of introduction, it is said that
“Motivated by these limitations, this work presents a comprehensive investigation of TTS for medical decision making tasks, introducing ‘a simple yet effective approach’ that enhances performance without additional supervision or specialized reward models.”

What is a simple yet effective approach? If you point at the two-stage TTS, make it clear by explicitly calling it out here. Also, this is a contribution of the paper. So, why not add it to the contribution list?

The contribution list does not need titles in italic in my opinion. For instance, “Comprehensive investigation" followed by “We systematically investigate …” sounds repetitive. The same goes for other items, too.

2.1.2 CoT Prompting
CoT requires more detailed steps than “Let’s think step by step.” I believe it deserves to be more well-defined and explained.

2.2 Scaling Test-time Compute
“General-purpose language models such as Llama or DeepSeek often struggle to provide accurate answers in complex medical tasks, and fine-tuning is prohibitively expensive due to the scarcity of expert-annotated reasoning data.” – Is this your finding? If not, it requires a reference, especially the first part regarding Llama and DeepSeek.

One-stage TTS:
“We estimate class probabilities by sampling N independent outputs from a large reasoning model under randomized decoding (e.g., temperature scaling (Guo et al., 2017)).” – What is the relevance of temperature scaling here? I know the paper and the method well. But the connection to randomized decoding is vague here. Did you try different temperature values to randomize outputs? Or what did you? Please, clarify. A brief note would help.

“which we map to a class via a parse” – parser or parsing function? Also with white space before the equation.

3.1 Datasets and Models

“To further assess generalizability across modalities and disease types in medical image diagnosis, we uses” – we use

Table 2: shows results with two-stage TTS with VLM. N=16. What about M? Eqs. 5 and 6 indicate that two-stage TTS requires an inner loop, right? In this respect, if M != 1, it would be also good to discuss the impact of inner loop on performance (both computation and modeling efficacy). This might incur a grid search but it is what it is.

3.3 Scaling Laws for Test-time Compute
Figure 2a shows that the performance saturates at some point in the textual domain. However, Fig. 2b and c indicates that it might further increase with N. Why did you stop there? What is the limit of N (# of samples)? If combined with M, as in my previous comment, these results could be much more interesting.

**Justification Of Final Rating:**

The authors have addressed my comments well. I understand their computational constraints at the moment so I won't push back on scaling N beyond 16. I also thank them for reporting additional results regarding counterfactual reasoning for DR detection. This might be indeed an interesting avenue to pursue. A brief discussion could also help the current manuscript. Based on the authors feedback and my initial review, I believe this is a timely work with interesting results. I maintain my preliminary rating.

**Justification Of The Preliminary Rating:**

I believe the manuscript should appear as a full-blown article after reformatting and aforementioned changes, instead of a squeezed paper. It would make a much better contribution to the community that way. I will be happy to see the responses and adjust my rating accordingly.

**Questions To Address In The Rebuttal:**

In addition to those comments above, I will add the following.

I like the results from TTS, both in one-stage and two-stage cases, also that it outperforms CoT in certain settings despite its simplicity regarding annotation efforts. Now that you present results on referrable DR detection, a recent study comes to my mind: https://www.ophthalmologyscience.org/article/S2666-9145(25)00232-5/fulltext , which showed that asking the model to provide counterfactual reasoning improved performance. I believe this could be seen as a way to spend more time thinking at test-time. Could you couple TTS, especially the two-stage one, with counterfactual reasoning, at least for the DR experiment and see if it improves the performance? Or what do you think about counterfactual reasoning  if additional experiments are out of the question during rebuttal?

---

> ### Author Response · Authors · 2026-01-24
>
> We thank Reviewer xniB for the detailed and insightful feedback. Below, we address each point raised.
>
> ---
>
> ### W1 – Manuscript Structure
> > *"Crucial components buried into the appendix, especially the related work."*
>
> **Response:** Thank you for this observation. We will use the additional two pages allowed for the camera-ready version to incorporate discussions from this rebuttal and move content from the Appendix (related work and prompts) into the main manuscript.
>
> ---
>
> ### Q1 – "Simple Yet Effective Approach" Clarification
> > *"What is a simple yet effective approach? This should be explicit and added to the contribution list."*
>
> **Response:** We agree. The phrase originally referred to our reward-free TTS strategy. However, our two-stage framework is indeed a distinct contribution. We will revise the first bullet point to: *"We investigate inference strategies (direct answering and CoT) and introduce a two-stage reasoning framework for medical VQA, where a VLM produces textual descriptions that are aggregated by an LLM for diagnosis."*
>
> ---
>
> ### Q2 – Contribution List Formatting
> > *"Italic titles sound repetitive."*
>
> **Response:** Agreed. We will remove italic titles from the contribution list, retaining only the explanations.
>
> ---
>
> ### Q3 – CoT Prompting Definition
> > *"CoT requires more detailed explanation than 'Let's think step by step.'"*
>
> **Response:** We will provide a more detailed definition of CoT prompting by incorporating content from Appendix C.1.2 into the main manuscript.
>
> ---
>
> ### Q4 – Citation for Llama and DeepSeek Claims
> > *"Claims about Llama/DeepSeek struggling require references."*
>
> **Response:** Thank you for catching this. We will add proper citations including Berger et al. (2025) and Naliyatthaliyazchayil et al. (2025).
>
> ---
>
> ### Q5 – Temperature Scaling Clarification
> > *"What is the relevance of temperature scaling? Did you try different values?"*
>
> **Response:** Thank you for this clarification. We used temperature **sampling** (not scaling) with fixed T=0.7, following standard practices in reasoning tasks. The reference to Guo et al. (2017) was a terminology error and will be corrected with proper citations.
>
> ---
>
> ### Q6 – Typo and Whitespace
> **Response:** We will correct it to "parsing function" and fix the whitespace formatting.
>
> ---
>
> ### Q7 – Typo: "we uses"
> **Response:** Will be corrected to "we use."
>
> ---
>
> ### Q8 – Two-Stage TTS: Impact of Inner Loop (M)
> > *"What about M? If M ≠ 1, discuss the impact of the inner loop."*
>
> **Response:** We use M=1 in all experiments. In preliminary investigations, diagnosis $a^{(i,j)}$ remained unchanged for fixed description $v^{(i)}$, indicating predictive uncertainty originates from the reasoning (description) stage rather than decision-making stage. We found no measurable gain from scaling the inner loop. We will clarify this in the camera-ready version.
>
> ---
>
> ### Q9 – Scaling Laws: Why Stop at N=16?
> > *"Figures 2b-c suggest performance might increase further. Why stop at N=16?"*
>
> **Response:** We stopped at N=16 primarily due to computational constraints, as VLM inference is significantly more expensive than text-only models. We will acknowledge that larger N values may yield further improvements, subject to computational budget.
>
> ---
>
> ### Q10 – Counterfactual Reasoning for DR Detection
> > *"Could you couple TTS with counterfactual reasoning for the DR experiment?"*
>
> **Response:** Thank you for this suggestion. We conducted additional experiments incorporating counterfactual (CF) prompting in the second stage (diagnosis). Results: w/o TTS: 0.50 (baseline: 0.57), w/ TTS: 0.69 (baseline: 0.71). CF prompting yielded slightly lower performance. We conjecture that the primary error source stems from first-stage vision encoding, based on the recent literature showing VLMs often overlook critical visual information (Fu et al., 2025). This points to an exciting future direction: applying CF-based prompting at the first stage to reduce visual hallucinations and improve feature extraction quality, which could yield more substantial performance improvements than second-stage reasoning improvements alone.

---

> ### Comment · Reviewer_xniB · 2026-01-29
>
> Hi,
>
> Thanks for responding to my comments. I was wondering if you have an updated manuscript (a new .pdf) by now.
>
> I know it is not a must at this stage but I would love to read the whole thing again.
>
> Feel free to say NO if it is not available.
>
> Best,
>
> Your reviewer

---

> > ### Author Response · Authors · 2026-01-31
> >
> > We sincerely appreciate your time you have invested in reviewing our work.
> >
> > Unfortunately, the OpenReview system does not permit PDF uploads now. However, we want to assure you that we are fully committed to implementing all suggested revisions in the camera-ready version. Below is a detailed summary of major changes:
> >
> > ---
> >
> > * **Section 1 - Final paragraph & contribution list**: Explicitly identify our "simple yet effective approach" and revise the contribution list to remove italic formatting for improved readability
> >
> > > Motivated by these limitations, this work presents a comprehensive investigation of TTS for medical decision making tasks, introducing a framework that enhances performance without requiring additional supervision or specialized reward models. Our key contributions are summarized as follows:
> >
> > > 1. We investigate inference strategies (direct answering and CoT) and introduce a two-stage reasoning framework for medical decision making, where a VLM produces textual descriptions that are aggregated by an LLM for diagnosis.
> > > 2. We evaluate TTS on test-time inference strategies and show consistent improvements, with gains of up to 30.4 percentage points over single-pass baselines.
> > > 3. We present a characterization of TTS, deriving scaling laws that describe how performance improves with the number of samples and identifying conditions under which TTS yields reliable gains.
> >
> > * **Section 2.1.2 - CoT prompting definition**:
> >
> > > Chain-of-Thought (CoT) Prompting. Alternatively, to elicit explicit reasoning, we employ Chain-of-Thought (CoT) prompting. While we retain the same input prompt $q$ used in the Direct Answering method, we enforce a reasoning process by introducing the trigger phrase "Let's think step by step." as a prefix to the generation (Kojima et al., 2022).
> >
> > > By leveraging the instruction-tuned nature of these reasoning models, this prefix acts as a directive that strictly orders the output: the model first generates intermediate reasoning steps (e.g., detailed clinical observations) and subsequently produces the final answer $a$. This ensures that the diagnosis is grounded in the generated rationale rather than being a direct, unexplained prediction. We denote this approach as the one-stage CoT method.
> >
> > * **Section 2.2 - Temperature sampling**:
> >
> > > One-Stage TTS. We estimate class probabilities by sampling $N$ independent outputs from a large reasoning model under randomized decoding. Specifically, instead of greedy decoding which deterministically selects the next token with the highest probability, we use temperature sampling with $T=0.7$ (Wang et al., 2022). At each token position, the model's output logits $\mathbf{z} = (z_1, \ldots, z_V)$ are converted to a probability distribution via the softmax: $ p_i = \frac{\exp(z_i / T)}{\sum_{j=1}^{V} \exp(z_j / T)}$ where $V$ is the vocabulary size. The next token is then sampled from this distribution, allowing us to generate $N$ diverse reasoning paths.
> >
> > * **Section 2.2 - Eq. (5–6)**:
> >
> > > Two-Stage TTS.
> > For two-stage inference in medical image diagnosis, we can apply TTS both in the description stage and in the diagnosis stage. Formally, ... where $v^{(i)}$ denotes the $i$-th description sampled from the VLM in the first stage, and $a^{(i,j)}$ is the $j$-th diagnosis from the language model given that description in the second stage.
> >
> > > Empirically, we observe that even under randomized decoding, the diagnosis $a^{(i,j)}$ remains unchanged for a fixed description $v^{(i)}$. This indicates that the predictive uncertainty originates from the reasoning process (description stage) rather than from the decision-making process (diagnosis stage). Consequently, there is no measurable gain from scaling test-time compute in the second stage, and we therefore set $M=1$. The final class probabilities and prediction are then estimated in the same way as in the single-stage case.
> >
> > ---
> >
> > We will also address the remaining comments in the final version.

---

### Official Review · Reviewer_n6t7 · 2026-01-07

**Confidence:** 2
**Preliminary Rating:** 4
**Final Rating:** 4

**Summary:**

This paper presents a comprehensive empirical and analytical investigation into test-time scaling for clinical decision-making, proposing a reward-free approach that utilizes self-consistency decoding across direct answering, chain-of-thought, and a novel two-stage multimodal reasoning framework. The authors demonstrate that allocating additional inference-time computation yields consistent performance improvements of up to 30 percentage points on both text-based medical benchmarks and medical imaging tasks without the need for supervised fine-tuning. Additionally, the study derives theoretical scaling laws to characterize these improvements, establishing that test-time scaling is most effective when the underlying model possesses a non-trivial baseline competence.

**Strengths:**

1. The study provides extensive empirical validation across diverse medical modalities, demonstrating consistent performance gains of up to 30 percentage points over single-pass baselines.
2. The paper offers a strong theoretical foundation by deriving scaling laws that analytically characterize the relationship between sample size and prediction reliability.
3. The proposed reward-free inference strategy effectively bypasses the need for costly medical data annotation and fine-tuning while achieving robust results.

**Weaknesses:**

1. The analysis lacks a discussion on the significant computational overhead and increased latency introduced by sampling, which are critical factors for deployment in time-sensitive clinical settings.

2. The study's conclusions are drawn from evaluations on benchmark datasets with standardized, relatively low-resolution images, which may not fully capture the complexity and variability of real-world clinical data.

3. The paper's framing of "clinical decision making" is limited to narrow classification and question-answering tasks, not addressing the more complex, multi-modal reasoning required in actual diagnostic workflows.

4. A qualitative error analysis is absent, which would be crucial for understanding the specific failure modes of the method and for building trust in its application for high-stakes medical decisions.

5. The investigation primarily uses moderately-sized models, leaving the effectiveness of test-time scaling on larger, more capable state-of-the-art foundation models as an open question.

**Detailed Comments:**

Please refer to the Weaknesses.

**Justification Of Final Rating:**

Based on my thorough review of the paper, along with careful consideration of the detailed comments provided by other reviewers and the thoughtful responses from the authors addressing each of their concerns, I have given the matter significant reflection and have ultimately decided to uphold my preliminary rating of weak accept for this submission.

**Justification Of The Preliminary Rating:**

I recommend a Weak Accept for this work because it provides a timely and comprehensive investigation into test-time scaling for medical AI, supported by strong empirical gains and solid theoretical backing, although the evaluation would be significantly strengthened by addressing the practical implications of increased inference latency and testing on more complex, non-standardized clinical datasets to prove real-world viability.

**Questions To Address In The Rebuttal:**

Please refer to the Weaknesses.

---

> ### Author Response · Authors · 2026-01-24
>
> We thank Reviewer n6t7 for the positive evaluation and valuable feedback. Below, we address each concern raised.
>
> ---
>
> ### W1 – Computational Overhead and Latency
>
> > *"The analysis lacks a discussion on the significant computational overhead and increased latency introduced by sampling, which are critical factors for deployment in time-sensitive clinical settings."*
>
> **Response:** Our inference cost comprises three components: (1) prompt processing: O(1) via KV caching as prompts are processed once and reused, (2) generation: O(N), and (3) self-consistency aggregation: O(N) but negligible as it runs on CPU. **Total cost scales linearly with N** (number of samples). Therefore, Figure 2 directly visualizes the cost-performance trade-off, where the x-axis represents computational cost.
>
> Importantly, latency need not scale linearly with N in practice, since samples can be generated simultaneously across multiple GPUs through parallel inference. We will add explicit discussion of deployment considerations and latency management strategies in the revised manuscript.
>
> ---
>
> ### W2 and W3 – Limited Scopes
>
> > *"The study's conclusions are drawn from evaluations on benchmark datasets with standardized, relatively low-resolution images, which may not fully capture the complexity and variability of real-world clinical data."*
>
> > *"The paper's framing of 'clinical decision making' is limited to narrow classification and question-answering tasks, not addressing the more complex, multi-modal reasoning required in actual diagnostic workflows."*
>
> **Response:** We thank the reviewer for these thoughtful comments. We agree that validation on full-resolution clinical data and evaluation on more comprehensive diagnostic scenarios would strengthen the generalizability of our approach. For future work, we plan to evaluate on more heterogeneous datasets with diverse resolutions and imaging protocols under more realistic clinical scenarios. We will clarify these limitations and future directions in the revised manuscript.
>
> ---
>
> ### W4 – Absence of Qualitative Error Analysis
>
> > *"A qualitative error analysis is absent, which would be crucial for understanding the specific failure modes of the method and for building trust in its application for high-stakes medical decisions."*
>
> **Response:** We thank the reviewer for this insightful suggestion. While our current manuscript focuses on quantitative evaluation, we have conducted preliminary investigations into failure modes through systematic ablation studies of second-stage reasoning models and prompting strategies. These experiments revealed that final performance remained comparable across different reasoning approaches.
>
> These findings suggest that the primary error source in our two-stage pipeline stems from first-stage vision encoding rather than second-stage reasoning: a conclusion consistent with recent literature showing that VLMs often overlook critical information in their visual representations [1].
>
> We will include a dedicated qualitative error analysis in the camera-ready version. We plan to employ attention visualization and Grad-CAM analysis on the vision encoder to examine whether models attend to diagnostically relevant image regions, comparing attention patterns between correct and incorrect predictions. This analysis will provide interpretable insights into specific failure modes and help build trust for clinical applications. We welcome additional suggestions from the reviewer.
>
> ---
>
> ### W5 – Limited to Moderately-Sized Models
>
> > *"The investigation primarily uses moderately-sized models, leaving the effectiveness of test-time scaling on larger, more capable state-of-the-art foundation models as an open question."*
>
> **Response:** We acknowledge that computational constraints limited our evaluation to moderately-sized models. Kim et al. [2] showed consistent improvements when models exceed 4B parameters. Similarly, our ablation studies show that model capacity matters for self-consistency decoding TTS. Given these findings, we hypothesize that applying TTS to larger foundation models would yield improvements comparable to or exceeding those observed in medium-sized models, as larger models' enhanced reasoning capabilities would likely produce more diverse and higher-quality candidate outputs for aggregation.
>
> However, conducting extensive experiments on larger models (e.g., 70B+ parameters) was not feasible within our computational budget. We will clarify this limitation and discuss expected scaling behavior with larger models in the revised manuscript.
>
> ---
>
> **References:**
>
> - [1] Fu, Stephanie, et al. "Hidden in plain sight: VLMs overlook their visual representations." arXiv preprint arXiv:2506.08008 (2025).
>
> - [2] Kim, Junhyuck, et al. "Not All Bits Are Equal: Scale-Dependent Memory Optimization Strategies for Reasoning Models." arXiv preprint arXiv:2510.10964 (2025).

---

### Official Review · Reviewer_yPrB · 2026-01-10

**Confidence:** 4
**Preliminary Rating:** 3
**Final Rating:** 4

**Summary:**

This paper investigates test-time scaling (TTS) for clinical decision-making tasks, applying self-consistency decoding to both text-based medical QA and medical image diagnosis. The authors evaluate three inference strategies (direct answering, chain-of-thought, and two-stage reasoning) across MMLU medical domains and MedMNIST datasets, demonstrating performance improvements. They also provide theoretical analysis characterizing when TTS is effective.

**Strengths:**

1. Experimental design is comprehensive. The paper systematically evaluates TTS across different inference strategies, models, modalities and medical tasks.
2. The theoretical framework provides useful analytical bounds showing when majority voting helps/hurts, which is an insightful addition to support the empirical findings.

**Weaknesses:**

1. Only tested one TTS methods, which is quite limited. How about other TTS methods for example best of N?
2. There is no discussion of the possible inference cost trade-offs which is crucial for clinical deployment.
3. How do these findings generalize to medical-specific VLMs such as LLaVA-Med? Also more challenging medical reasoning benchmarks such as MedQA and PubMedQA?

**Detailed Comments:**

1. Two-stage reasoning seems to be another way of dissecting the models. How does the best model compare to some naive baselines, for example just using a vision encoder to encode the features and train a simple classifier such as SVM to do the classification?

**Justification Of Final Rating:**

I thank the authors for their additional evaluations on challenging benchmarks and models based on my suggestions. Given the promising results and the reasonable discussion, I would like to increase my score.

**Justification Of The Preliminary Rating:**

The paper provides a thorough empirical investigation of an important practical problem. While the technical novelty is limited, the systematic evaluation across modalities and model sizes and the theoretical justification are comprehensive.

**Questions To Address In The Rebuttal:**

See above.

---

> ### Author Response · Authors · 2026-01-24
>
> We thank Reviewer yPrB for the thoughtful review and constructive feedback. Below, we address each concern raised.
>
> ---
>
> ### W1 – Limited TTS Methods
>
> > *"Only tested one TTS methods, which is quite limited. How about other TTS methods for example best of N?"*
>
> **Response:** We thank the reviewer for the thoughtful comment. As noted in the Introduction, directly transferring verifier-based TTS methods (best-of-N, beam search) to medical applications is not feasible due to the absence of reliable verifiers. Training reward models requires vast labeled data (e.g., Qwen-PRM used 4.5M labels). While LLM-as-a-judge schemes exist, they depend on the judge's capability rather than the original VLM.
>
> Critically, we observed that pass@64 (probability that there exists at least one correct answer in 64 samples) exceeds 90% across datasets and models. This demonstrates that a perfect verifier could achieve 90+% accuracy even with 1B models, highlighting the urgent need for reliable reward models in medical AI. Existing verifier work [1,2] is domain-specific (e.g., text-only) and inapplicable to multimodal radiology. Hence, we focused on reward-free schemes like self-consistency.
>
> We will include these discussions in the final draft.
>
> **References:**
>
> - [1] Thomas, Alois, et al. "Process Reward Models for Sentence-Level Verification of LVLM Radiology Reports." *arXiv preprint arXiv:2510.23217* (2025).
>
> - [2] Wang, Hanyin, et al. "Process-supervised reward models for verifying clinical note generation: A scalable approach guided by domain expertise." *Proceedings of the 2025 Conference on Empirical Methods in Natural Processing.* 2025.
>
> ---
>
> ### W2 – Inference Cost Trade-offs
>
> > *"There is no discussion of the possible inference cost trade-offs which is crucial for clinical deployment."*
>
> **Response:** We thank the reviewer for this important observation. Inference cost is dominated by generation (scales linearly with $N$), as prompt processing (cached to $O(1)$) and voting (negligible CPU overhead) are minor. Therefore, Figure 2 directly visualizes the cost-performance trade-off, where the x-axis represents computational cost. Note that parallelization on GPUs can further mitigate latency. We will add this discussion to Sec 3.3.
>
> ---
>
> ### W3 – Generalization to Medical-Specific VLMs and Challenging Benchmarks
>
> > *"How do these findings generalize to medical-specific VLMs such as LLaVA-Med? Also more challenging medical reasoning benchmarks such as MedQA and PubMedQA?"*
>
> **Response:** We added MedQA analysis (Table W3). Overall, we observe the same trend supporting our paper's findings. Performance improves monotonically with $N$ across most models and prompt strategies, confirming the effectiveness of self-consistency decoding. For the 1B + CoT approach, we observe performance degradation with TTS, which is consistent with our findings in the paper. When the base model's reasoning is unreliable (as evidenced by the low $N=1$ accuracy of 16.11%), self-consistency amplifies incorrect patterns rather than correcting them, leading to decreased performance as $N$ increases.
>
> **Table W3. MedQA with Self-Consistency TTS**
>
> | Model | Prompt | $N=1$ | $N=4$ | $N=16$ | $N=64$ |
> |:------|:-------|:------|:------|:-------|:-------|
> | Llama-8B | Zero | 58.0±4.7 | 60.3±0.8 | 61.7±0.5 | **61.9±0.3** |
> | Llama-8B | CoT | 62.7±1.5 | 69.9±0.9 | 74.6±0.6 | **75.6±0.3** |
> | Llama-3B | Zero | 47.7±4.9 | 50.1±0.8 | 51.2±0.6 | **51.9±0.3** |
> | Llama-3B | CoT | 37.9±2.3 | 42.9±1.0 | 49.6±0.9 | **52.9±0.5** |
> | Llama-1B | Zero | 29.7±3.4 | 31.3±1.1 | 33.4±0.8 | **34.8±0.5** |
> | Llama-1B | CoT | **16.1±1.3** | 12.6±1.0 | 6.4±0.5 | 3.1±0.3 |
> | DeepSeek-8B | Zero | 37.3±5.0 | 40.1±0.9 | 42.3±0.6 | **42.7±0.4** |
> | DeepSeek-8B | CoT | 50.2±1.1 | 55.1±0.9 | 57.6±0.7 | **58.2±0.4** |
>
> Due to the short rebuttal period and our limited GPU resources, we could not run our methods with LLaVA models on vision tasks, but we promise to add additional analysis on those models for more complex VQA tasks such as VQA-RAD to the final camera-ready version.
>
> ---
>
> ### Q1 – Comparison to Naive Baselines
>
> > *"Two-stage reasoning seems to be another way of dissecting the models. How does the best model compare to some naive baselines, for example just using a vision encoder to encode the features and train a simple classifier such as SVM to do the classification?"*
>
> **Response:** We appreciate this question. Our work does not claim SOTA performance but rather demonstrates *TTS utility in a training-free setting*. Vision encoder + SVM requires labeled data and training, whereas our approach leverages pretrained VLM reasoning without parameter updates. That said, we acknowledge exploring how our two-stage reasoning framework could potentially be adapted to or compared with traditional classifiers is a compelling direction for future research.

---

> > ### Author Response · Authors · 2026-01-30
> >
> > We sincerely thank the reviewer again for your time and effort, along with valuable feedback.
> >
> > Following your suggestion, we conducted additional experiments using **LLaVA-Med-v1.5-Mistral-7B** on two popular and challenging medical VQA benchmarks: SLAKE and RAD-VQA. The results are summarized below:
> >
> > | Model | Method | SLAKE | RAD-VQA |
> > |:------|:-------|:----------|:------------|
> > | LLaVA-Med | Direct Answering | 0.52 | 0.59 |
> > | | Direct Answering (+TTS) | 0.53 (↑ +0.9pp) | 0.58 (↓ -0.8pp) |
> > | | One-stage CoT | 0.54 | 0.54 |
> > | | One-stage CoT (+TTS) | 0.55 (↑ +1.7pp) | **0.59** (↑ +5.6pp) |
> > | | Two-stage Reasoning | 0.65 | 0.57 |
> > | | Two-stage Reasoning (+TTS) | **0.66** (↑ +2.0pp) | **0.59** (↑ +2.0pp) |
> >
> > Our findings on LLaVA-Med are consistent with our main results, demonstrating that: (1) CoT-based methods outperform Direct Answering, and Two-stage Reasoning achieves the best overall performance; (2) Two-stage Reasoning provides substantial improvements over Direct Answering, particularly on SLAKE (+13pp). These results suggest that our proposed approach generalizes well to domain-specific medical VLMs.
> >
> > We will incorporate these results and discussion into the revised paper. We hope this addresses your concern. Please let us know if you have any further questions or comments. We are happy to provide additional clarifications. Thank you again for helping us strengthen the contribution of our work.

---

> ### Author Response · Authors · 2026-01-31
>
> Dear ACs,
>
> Thank you for your time, service, and invaluable contributions to the MIDL community.
>
> We have noticed that reviewer yPrB has not yet engaged in the discussion phase, and we believe their input is crucial for a fair evaluation. We would be very grateful if you could kindly encourage the reviewer to start the discussion.
>
> Thank you again for your time and for serving as AC for this conference. We truly appreciate it.

---

> ### Comment · Area_Chair_n8MX · 2026-02-01
> **Urgent: Please provide your updated score on the paper**
>
> Dear Reviewer yPrB,
>
> Could you please review the author's rebuttal? As we approach the end of the discussion period, I kindly ask you to update your Final Rating by going to Edit → Official Review and entering your final score by February 1st, 2026 (23:59 AoE).

---

### Comment · Area_Chair_n8MX · 2026-02-01
**Reminder: Final Rating Update Requested (Feb 1, AoE)**

Dear Reviewers,

Thank you for your reviews and participation in the discussion. As we are approaching the end of the discussion period, I kindly ask you to please update your Final Rating by going to Edit → Official Review and entering your final score by February 1st, 2026 (23:59 AoE).

If applicable, please also consider briefly responding to the authors’ rebuttal and engaging with other reviewers’ comments, especially where there are differing opinions. This will be very helpful for the meta-review and final decision process.

Thank you again for your time and contribution.

---

### Meta-Review · Area_Chair_n8MX · 2026-02-08

**Recommendation:** Accept (Poster)
**Confidence:** 4

**Metareview:**

There is clear consensus that this paper provides a thorough and technically sound empirical study of test-time scaling for medical decision-making. The authors effectively addressed concerns regarding inference costs and benchmarks during the rebuttal, confirming the work's value for the MIDL community.

---

### Decision · Program_Chairs · 2026-02-13

Accept (Poster)